# Neuroadaptive Biochemical Mechanisms of Remote Ischemic Conditioning

**DOI:** 10.3390/ijms242317032

**Published:** 2023-12-01

**Authors:** Ksenia Baranova, Natalia Nalivaeva, Elena Rybnikova

**Affiliations:** I. P. Pavlov Institute of Physiology, Russian Academy of Sciences, 199034 Saint Petersburg, Russia; ksentippa@mail.ru (K.B.); natalia.nalivaeva@outlook.com (N.N.)

**Keywords:** ischemic conditioning, hippocampus, brain, adaptation, mechanisms of neuroprotection, hypoxia-inducible factor-1 (HIF-1), hypothalamic–pituitary–adrenocortical axis (HPA)

## Abstract

This review summarizes the currently known biochemical neuroadaptive mechanisms of remote ischemic conditioning. In particular, it focuses on the significance of the pro-adaptive effects of remote ischemic conditioning which allow for the prevention of the neurological and cognitive impairments associated with hippocampal dysregulation after brain damage. The neuroimmunohumoral pathway transmitting a conditioning stimulus, as well as the molecular basis of the early and delayed phases of neuroprotection, including anti-apoptotic, anti-oxidant, and anti-inflammatory components, are also outlined. Based on the close interplay between the effects of ischemia, especially those mediated by interaction of hypoxia-inducible factors (HIFs) and steroid hormones, the involvement of the hypothalamic–pituitary–adrenocortical system in remote ischemic conditioning is also discussed.

## 1. Introduction

Ischemic preconditioning was first described in the heart as a reduction in the size of the myocardial infarction region provided by 5-min coronary artery occlusion cycles before a 40-min episode of ischemia [1]. It was then shown that brief episodes of occlusion–reperfusion of the circumflex coronary artery reduced damage caused by hourly occlusion of the adjacent descending artery [2]. Thus, a preconditioning can be remote, including intermittent occlusion of the abdominal aorta, the mesenteric, intestinal, and renal arteries, etc. [3,4]. Further, the concept of remote preconditioning was expanded by the application of short cycles of ischemia–reperfusion to the hind limb, demonstrating that intermittent ischemic impacts on an organ remote from the heart can provide significant cardioprotection [5]. Remote post-conditioning was first reported by Andreka and colleagues in a pig model of acute myocardial infarction resulting in a reduced infarct size when a 5-min episode of ischemia–reperfusion was applied to the hind limb after the end of coronary artery occlusion [6]. Further studies revealed the protective effect of remote ischemic conditioning (RIC) on other vital organs, including the brain [7], kidneys [8], liver, lungs, gastrointestinal tract, skin, etc. [4].

Based on numerous clinical and experimental data, it can be concluded that RIC, such as hypoxic or pharmacological conditioning, enhances the resistance of various organs to damaging effects not only of an ischemic nature; it also causes a systemic reaction involving universal endogenous adaptive mechanisms and increases the adaptive potential of the organism as a whole. For example, in liver resections and transplants [9], heart and coronary artery surgery [10], kidney surgery and reconstructive microsurgery [11], remote intermittent ischemia reduces overall operational stress, diastolic blood pressure, incidence of strokes, inflammation, edema, neurological and cognitive impairment, reduces days of treatment in the intensive care unit, and helps to increase cerebral blood flow and integrity of the blood–brain barrier. In the present review, we summarize some of the current literature and our knowledge and expertise in this fast-developing research area with a special focus on the hippocampus, which was the principal subject of the authors’ research for many years.

## 2. Effects of RIC on the Hippocampus

Stroke is one of the main causes of neurological disability worldwide, and the main therapeutic goals are to save the penumbra by restoring blood flow and reducing nerve cell death. The neuroprotective effects of RIC are primarily evaluated by the reduction of the size of cerebral infarction. Studies in rodents show that RIC either before, during, or after an ischemic episode, on average, can reduce infarct volume by 35–42% compared to controls depending on the animal models [12,13]. In mice, short cycles of application were shown to be more pronounced while in rats longer cycles were more effective [12,13]. While current approved therapies do not act directly on the brain parenchyma, RIC is effective not only against penumbra neuronal dysfunction, but has a positive effect on other brain regions and the brain as a whole, showing anti-apoptotic, anti-oxidant, and anti-inflammatory effects (for review, see [14]). For example, in a model of vascular cognitive impairment caused by total brain hypoperfusion a 2-week post-RIC resulted not only in an improvement of cerebral blood flow but also in a decrease in neurodegenerative signs (such as inflammation, cell death and beta-amyloid accumulation) in all areas of the brain [15]. It has been shown that, in rats, limb RIC before MCAO reduces whole-brain mitochondrial and total oxidative stress, lipid peroxidation, oxidative DNA damage, and oxidative protein damage [16].

The hippocampus is considered to be more sensitive to ischemia than other brain regions. Thus, in animal models of transient global ischemia, which mimics the delayed neuronal death caused by cerebral ischemic stroke, a significant increase in neuronal apoptosis was observed in the CA1 and CA2 regions of the hippocampus [17]. Several studies have shown that remote ischemic limb conditioning effectively reduces neuronal apoptosis in these areas of the hippocampus when applied shortly before an experimental stroke, as evidenced by DNA fragmentation and/or the formation of apoptotic bodies [18]. A very significant neuroprotective effect in the hippocampal region in rats was also observed with a single 20-min limb RIC applied as a very delayed post-RIC 2 days after a global 10-min ischemic stroke [19].

According to an interesting hypothesis, the hippocampus not only suffers from cerebral ischemia more than other regions but is also sensitive to any distant focal damage [20]. The development of post-stroke cognitive and psychopathological disorders is not directly related to the severity and localization of the primary brain lesion but is primarily due to the functional and structural changes in the hippocampus, a region of the brain that is selectively vulnerable to harmful external factors and responds to them with increased secretion of cytokines [21]. Moreover, ischemic, and focal brain lesions induce excessive secretion of stress corticosteroid hormones that interact with hippocampal receptors, triggering signaling pathways leading to neuroinflammation and subsequent impairment of neurogenesis, neurodegeneration, depression, and dementia [22,23].

Based on the sensitivity of the hippocampus to distant injury, one can also assume its susceptibility to remote conditioning. Indeed, employment of remote ischemic postconditioning for the treatment of experimental traumatic brain injury in mice allowed for the minimization of neuronal changes in the CA1 area of the hippocampus and improved cognitive functions and motor coordination [24,25]. A significant improvement in sensorimotor functions with the use of delayed post-RIC has also been shown in a model of neonatal hypoxia-ischemia in rats [26]. In a classical experimental model of focal cerebral ischemia–reperfusion, pre-RIC also improved spatial learning and memory capacity, probably due to its protection of cholinergic neurons in the CA1 region of the hippocampus [27]. After application of post-RIC in this model, a reduction in infarct area was observed when RIC was applied up to 3 h after stroke, but not when RIC was applied after 6 h or 2 months, although animal behavior dependent on the hippocampus integrity was improved in all cases [28]. In general, in experiments with occlusion of the middle cerebral or carotid artery, hypothermic circulatory arrest, etc. RIC significantly improved neurological, functional, cognitive, and behavioral parameters, in which the central role belongs to the hippocampus. Normalization of the functions of the hippocampus can explain the effectiveness of RIC in rehabilitation therapy, in the prevention and treatment of neurological diseases of ischemic or inflammatory origin, in traumatic brain injuries, vascular cognitive impairment, dementia and Alzheimer’s disease [13,29].

The effect of remote ischemia on the release of corticosteroids and the function of the hippocampus, as a regulator of various forms of behavioral plasticity, is in agreement with the anxiolytic and anti-depressant effects of limb ischemia–reperfusion treatment recently identified in our laboratory [30,31,32]. RIC has been found to normalize behavioral and hormonal outcomes in models of depression and PTSD (post-traumatic stress disorder), effectively preventing development of the disorders when applied in pre- and post-injury regimens [30,31,32]. It should be noted that in the paradigms of severe hypobaric hypoxia, RIC neither reduces the risk of altitude sickness in subjects, nor prevents acute mountain sickness, cerebral edema, or pulmonary edema, but facilitates rehabilitation [33]. In our experiments in rats, there was no increase in animal survival under the conditions of acute severe hypoxic exposure after RIC, but the rehabilitation period was significantly improved in the surviving animals subjected to RIC. In particular, they did not display the symptoms of post-hypoxic pathology related to the malfunctioning of the hippocampus.

## 3. RIC and Neuroimmunohumoral Pathway

The transmission of the protective ischemia–reperfusion effect from the limb to the target organ simultaneously involves the nervous, circulatory, and immune systems, working in close interaction. The somatosensory system, spinal cord, and autonomic nervous systems are involved in the neuronal pathway. The transmission there is carried out, according to the type of reflex, from sensory (afferent) neurons of the C-fibers to the integration center in the central nervous system and to the motor (efferent) neurons of the vagus nerve. This has been proven by experiments with nerve and spinal cord cutting [34,35,36] with the application of nicotinic and opioid receptor antagonists [37,38] or blockers of the afferent fiber, autonomic ganglia, and parasympathetic motor neurons [34,35,36,39], which canceled or reduced the protective effect of RIC, as well as by experiments in which electrical or chemical neurostimulation partially reproduced it [35,36,40]. Moreover, the afferent nerve pathway can be activated by physiologically active substances in the blood, and, vice versa, peripheral nerves can induce the synthesis of humoral factors. For example, activation of the vagus nerves, including indirect RIC, induces the synthesis and release of NO, nitrites, and glucagon-like peptide-1 (GLP-1) in visceral organs [41,42]. The non-selective TRPV1 cationic channels, expressed in the primary sensory nerves and regulated by physical or chemical stimuli, are activated during limb ischemia–reperfusion cycles and release neuropeptides, calcitonin gene-related peptide (CGRP), and substance P (SP) [43,44]. 

The release of biologically active substances into the bloodstream in response to RIC has also been demonstrated in various models. Today, among the main humoral factors for the realization of the protective effect of remote ischemia are listed adenosine, nitrite/NO, CGRP, bradykinin, catecholamines (norepinephrine), microRNAs, endogenous opioids, free radicals, cytokines (IL-1α, IL-10) and chemokines (stromal factor 1α (SDF-1α)), GLP-1, apolipoprotein AI, hydrophobic peptides, prostanoids, endocannabinoids, leukotrienes, adrenomedullin, etc. [42,45,46]. It is assumed that some of them are transferred to the target organ by endogenous extracellular vesicular nanoparticles originating from the endothelium, hematopoietic cells, or platelets [47]. Exosomes interact with target cells using a number of surface molecules. Because exosomes carry chemokines and microRNAs they can, via HIF-1α, reduce levels of IL-6 and tumor necrosis factor-α (TNFα), and via vascular endothelial growth factor VEGF and NO synthases mediate angiogenesis, while via Hsp70 and Bim they inhibit apoptosis in target cells [48,49]. Many of the humoral factors (norepinephrine, biologically active peptides, NO, opioids, etc.) are able to bind to receptors in the nervous system, acting as neurotransmitters or modulators. One such example is the interaction of GLP-1 with sensory fibers [50]. In plasma samples of patients exposed to RIC, an increase in the concentration of biogenic amines and amino acids, in particular glycine, which is an inhibitory neurotransmitter, was reported [51]. 

It has recently been shown that parasympathetic pathways stimulated by RIC provide organ protection not only by direct innervation, but also through immunomodulation, inhibiting cytokine production and integrin expression on neutrophils via the cholinergic anti-inflammatory pathway, but only in the presence of an intact and innervated spleen. RIC also reduces inflammation through the release of anti-inflammatory exosomes, affects circulating white blood cells and immune precursors in the spleen [14,44,48,52].

RIC protective effects at the organism level also involve augmentation of blood circulation [53,54], remodeling in the organs and tissues [55] and changes in the systemic immune and inflammatory response [56]. There are studies suggesting that RIC applications improve microcirculation in various organs under normal and pathological conditions [57,58,59]. Although it is technically difficult to assess the effect of RIC on brain microcirculation, there is evidence that multiple RIC applications improve cerebral blood flow [60] and enhance dynamic cerebral autoregulation in healthy individuals [61], which requires further study of its effects on brain vasculature and endothelial remodeling. Nevertheless, it is already evident that RIC in pre- and postconditioning modes is protective against brain cerebrovascular cognitive dysfunction [62]. The possible molecular mechanisms of RIC effects on endothelial and neuronal cells will be discussed in subsequent chapters.

## 4. Phases of RIC Neuroprotection

For RIC, as for other types of conditioning effects, two phases of neuroprotection are characteristic (Figure 1). The early phase begins immediately after the conditioning stimulus and lasts for several hours, while the late phase occurs after 12–48 h and persists for several days and even months (the “second window”) [63,64]. The signals of alternating cycles of ischemia–reperfusion or occlusion–deocclusion enter the brain via the neuroimmunohumoral pathway in the form of extracellular primary messengers (neurotransmitters, hormones, neuromodulators, cytokines, and other RIC factors), initiating molecular events at the level of membrane receptors and ion channels [65]. Neuronal receptors, having received a signal, activate the calcium, phosphoinositide, and cyclic nucleotide systems of intracellular signal transduction in the form of corresponding secondary intracellular messengers (cAMP or cGMP, Ca^2+^, inositol trisphosphate (PI3), NO, etc.), which activate or inhibit effector protein kinases and proteolytic enzymes, which, in turn, modulate functionally important proteins of ion channels, receptors, synaptic vesicles, cytoskeleton, mitochondria, etc. These processes form the basis of an early wave of adaptive responses [66,67,68].

The signal from the ischemic limb can stimulate G-protein-bound receptors on the surface of nerve cells, which is accompanied by activation of the protein kinases type C, type B (Akt), cytoplasmic tyrosine kinase (Trk), and MAP kinases. Kinases C and B are inhibited by phosphorylation of glycogen synthetase kinase-3β (GSK3β), which is one of the key mechanisms for keeping nonspecific mitochondrial pores in a closed state, preventing loss of proton gradient and separation of oxidation and phosphorylation, organelle damage, intracellular edema, calcium overload, and apoptosis. The cardio- and neuroprotective effects of ischemic conditioning are also mediated via phosphorylation by serine and threonine kinases of ATP-dependent potassium channels of sarcolemma and mitochondrial membranes leading to their opening. This prevents the formation of mitochondrial pores and is associated with an increase in the generation of superoxide radicals in the respiratory chain [69,70]. Bradykinin, binding to B2 receptors (BDKRB2), participates in remote conditioning as an endogenous cytoprotective mediator, providing the triggering of the mitochondrial anti-apoptotic pathway through activation of PI3K/Akt/eNOS signaling and regulation of redox status through the release of NO [66,67]. Rapid RIC-mediated energy-saving effects are also due to the activation and then increase in the level of adenosine monophosphate-activated protein kinase (AMPK), a regulator of cellular energy homeostasis, which can be activated, for example, by a factor inhibiting macrophage migration (MIF), the level of which rises in plasma after ischemia [71,72]. In addition, the formation of neuronal tolerance after conditioning involves the sequential activation of the VEGF receptor, which has tyrosine kinase activity [73].

In addition to phosphorylation, nitrosylation of key mitochondrial proteins mediates the rapid cytoprotective effect of nitrite (humoral factor RIC) by reducing the generation of mitochondrial reactive oxygen species (ROS), which contributes to the preservation of the closed pore of mitochondria and inhibits the release of cytochrome c [74]. Acetylcholine provides a rapid anti-inflammatory effect through nicotinic receptors, by which it inhibits the release of TNFα from macrophages and inhibits the polymerization of F-actin, which is critical for the expression on the surface of circulating neutrophils of β2-integrin CD11b, which regulates their adhesion and migration [74].

Inhibition and suppression of expression, as a result of RIC, of the water channel protein aquaporin (AQP), located in the terminal legs of astrocytes, prevents an increase in transmembrane water flow, cytotoxic edema of astrocytes, and an increase in the permeability of the BBB. There is also a general improvement in functional neurological recovery after stroke due to the suppression of AQP4 in astrocytes [75]. Another molecular target of remote post-conditioning is intracellular matrix metallopeptidase 9 (MMP-9), which destroys the components of dense contacts between endothelial cells [76].

The “second window” of protection, as opposed to the early phase, depends on gene expression and protein synthesis, which are triggered by kinases through the induction of transcription factors to form the long-term stability of brain neurons [64,77]. Some proteins may be involved in both phases of neuroprotection, e.g., intracellular messenger and transcription factor STAT3 (signal transmitter and transcription activator 3). Activation of membrane receptors gp130, TNFR2, and S1PR in the nervous system by ligands such as IL-6 and 10, sphingosine-1-phosphate S1P, TNFα, lipoproteins, melatonin, erythropoietin (EPO), insulin, and leptin lead to phosphorylation, mainly by JAK kinase, of the STAT3 factor. Activated by the conditioning signal, P-STAT3 is dimerized and moved to the nucleus, where it regulates the transcription of the genes of anti-oxidant, anti-apoptotic, and proangiogenic protective proteins Bcl-xl, cytochrome c oxidase subunit 2 (COX2), cytokine signalling suppressor 3 (SOCS3), MCL-1 (pro-survival factor myeloid cell leukemia-1), superoxide dismutase-2 (SOD2), VEGF, metallothionein, etc. As a result, it reduces neuronal inflammation and apoptosis. STAT3 is not only a transcription factor but also non-genomically regulates the functions of mitochondria, endoplasmic reticulum, and lysosomes. Phosphorylation and other modifications allow STAT3 to interact with GRIM19 (a component of mitochondrial complex I) and TOM20 (translocase of the outer membrane) to enter the mitochondria. There, STAT3 interacts with several proteins, promoting Ca^2+^ homeostasis, maintaining electron transport chain activity, increasing ATP levels, and reducing ROS production by inhibiting mitochondrial pore opening [78].

## 5. Delayed Anti-Apoptotic Mechanisms Induced by RIC

The mechanisms by which remote conditioning reduces cerebral infarction include, first of all, inhibition of cellular apoptosis, both internal, mediated by mitochondria, and external, triggered by death receptors [79,80] (Figure 1 and Figure 2).

The main intracellular signaling anti-apoptotic pathways activated by RIC and mediating its neuroprotective effects are the Akt, mTOR, MAPK, PKC, and TLR4 pathways [81]. The PI3K/Akt pathway is an important signaling mediator that regulates cell survival by inhibiting the processes of apoptosis [82]. The mTOR signaling pathway plays a central role in cellular metabolism, differentiation, development, autophagy, and survival [83]. The MAPK family is a major family of regulatory kinases that transform extracellular signals into cellular responses, participating in many physiological and pathological processes and playing a certain role in the mechanisms of RIC [84]. PKC is a family of serine/threonine protein kinases, of which δPKC and εPKC counter-balance cerebral damage. Remote ischemic pre- and post-conditioning via the endogenous ROS-dependent signaling cascade induces cleavage of δPKC the activity of which usually contributes to cell death but induces phosphorylation of εPKC, which promotes the survival of neurons. RIC also inhibits expression of toll-like receptor 4 (TLR4), which is an important mediator of the innate immune response, mediates neuroinflammation and is involved in ischemic tolerance. RIC in pre- and postconditioning mode via the TLR4/MyD88 signaling pathway was shown to attenuate ischemic brain injury and improve neurobehavioral function in aged rats [85].

The RIC stimulus also inhibits the translocation of the poly (ADP-ribose) polymerase (PARP) from the nucleus to the mitochondria, which prevents the release of mitochondrial apoptosis-inducing factor (AIF), an electron transport flavoprotein that itself plays an important role in the survival and death of neuronal cells. In turn, it activates the enzyme poly (ADP-ribose) polymerase-1 (PARP-1), which is also responsible for various neurological disorders. In addition, remote ischemia–reperfusion blocks the physical interaction of nuclear AIF with cyclophilin A and histone H2AX, which, in combination with inhibition of AIF/PARP pathways, prevents chromatin condensation and DNA fragmentation of neuronal cells [86].

Another mechanism by which RIC attenuates cerebral damage in animal models involves suppression of the tumor necrosis factor–related apoptosis-inducing ligand (TRAIL). It prevents TRAIL released by glia from binding to death receptors (DR) to form a death-inducing signaling complex (DISC) that triggers neuronal apoptosis through activation of the caspase-8/caspase-3 pathway [87]. Moreover, limb ischemia inhibits neuronal apoptosis by directly attenuating the activation of caspase-3 and TRAIL death receptors [88]. In addition, RIC enhances mRNA expression of the anti-apoptotic inhibitory protein cFLIP, which, as a close isoform of caspase-8, can instead physically bind to the catalytic site of the FAS-associated death domain (FADD), thereby preventing formation of DISC and suppressing the transmission of apoptotic signals [77].

Cycles of limb RIC were also shown to promote neuronal survival by reducing in the brain the content of ischemia–reperfusion-induced lipocalin Lcn-2, a protein that regulates inflammation, iron metabolism, and cell death, and decreasing the number of Lcn-2-positive astrocytes [89]. This prevents the interaction of Lcn-2 with its receptor (carrier of organic cations) on the membranes of neurons. This reduces neuronal expression of the pro-apoptotic Bcl-2 family, in particular, of the proapoptotic gene *Bim*, an inducer of post-ischemic neuronal death [90]. Moreover, when using ischemic conditioning, a decrease in the expression of pro-apoptotic proteins Bax, Bid, and caspase-3 and an increase in anti-apoptotic Bcl-2 and Bcl-xl were demonstrated, which inhibits the opening of the mitochondrial pore and effectively reduces apoptosis [91].

Brain-derived neurotrophic factor BDNF is another likely mediator of RIC-induced neuroprotection. BDNF binds to tropomyosin receptor kinase B (TrkB receptor), mobilizing TrkB kinase to activate the MAPK/ERK and PI3K/Akt signaling cascades that promote neuronal differentiation and survival. BDNF induced by ROS activates HIF-1α, the nuclear factor erythroid (Nrf2) and their target gene programs. Nrf2, in turn, can initiate BDNF expression [92]. The PI3K/Akt signaling pathway was also shown to regulate expression of murine E3-ubiquitin ligase MDM2, which, when phosphorylated, stabilizes in the nucleus and interacts with p53, which leads to destabilization of p53, thus preventing neuronal apoptosis [93].

The prevention of neuronal damage can also be achieved by Inducing the mechanism of microautophagy, which allows for the removal of dysfunctional cellular components, especially of damaged mitochondria, preventing the release of cytochrome c and the transmission of death signals. Remote ischemia affects the expression of checkpoint proteins of cellular autophagy/apoptosis. For example, it reduces plasma levels of HMGB1, a protein secreted by immune cells as a cytokine mediator of inflammation, but stimulates cytosolic HMGB1, which regulates apoptosis, protecting the autophagy-related proteins beclin-1 and ATG5 from calpain-mediated cleavage [94,95]. Initiation of the opioid receptor/PI3K/AKT/GSK3β signaling pathway under RIC can also lead to phosphorylation of Bcl-2 and breakdown of the Bcl-2/Beclin-1 complex, which plays an important role in stimulating autophagy and reducing mitochondrial damage in conditioned rats after cerebral ischemia [96]. Moreover, when using RIC, there is a predominance of the conjugated form of phosphatidylethanolamine with the microtubule-bound protein LC3-II, which is recruited to the membranes of the autophagosome. This leads to a decrease in sequestosome 1 levels and a suppression of mTOR kinase, as well as an increase in beclin-1 and hemoxygenase-1 (HO1), which mediate pro-autophagic signaling and prevention of cell death [96,97,98] (Figure 1 and Figure 2).

## 6. Anti-Oxidant and Anti-Inflammatory Mechanisms of RIC

One of the main, closely interrelated mechanisms for preventing neurological dysfunction after focal cerebral injuries by RIC is the reduction of oxidative stress in neurons. It can be detected at the level of malondialdehyde and 8-hydroxy-2-deoxyguanosine, as well as of neuroinflammation at the levels of myeloperoxidase, TNFα, IL-1 and IL-6 [99,100,101] (Figure 1 and Figure 2).

Postischemic neuroinflammation is known to cause an imbalance between oxidative stress and anti-oxidant systems, and excessive accumulation of free radicals leads to oxidative damage to proteins, lipids, and nucleic acids. As such, oxidative stress contributes significantly to the progression of ischemic neuronal insufficiency after cerebral damage. RIC prevents these disorders primarily by activating anti-oxidant systems and triggering the expression of proteins such as Nrf2 and HO1, quinone oxidoreductase 1 (NQO1), and superoxide dismutase (SOD). The use of RIC contributes to the movement of the transcription factor Nrf2 from the cytosol to the nucleus, where it binds to the DNA promoter anti-oxidant response element ARE. This initiates the transcription of anti-oxidant cytoprotective proteins and enzymes, i.e., the key components of the anti-oxidant systems of glutathione and thioredoxine and the enzymes regenerating NADP levels [102]. Those, in turn, mediate a significant decrease in the level of nitrotyrosine, P22 (catalytic subunit of NADPH oxidase) mRNA, and xanthine oxidase, which play a fundamental role in maintaining redox homeostasis [103]. Secondly, RIC reduces the production of superoxide by reversing the activity of endothelial NO synthase (eNOS). This enzyme, in the presence of calmodulin cofactors—in particular, of tetrahydrobiopterin (BH_4_)—produces mainly NO, but with their lack or decrease in the affinity, the “uncoupled” eNOS produces mainly superoxide [104]. Regulation of eNOS is also important for hemodynamics, as NO is a potent vasodilator and improves microcirculation and counteracts reperfusion damage by reducing ROS production. Moderate NO formation during RIC induces cerebroprotective adaptations. Moreover, through the PI3K/Akt signaling pathway in astrocytes, it activates the synthesis of the glutamate-1 transporter (GLT-1), which removes glutamate from the synaptic cleft and prevents excitotoxicity [105,106].

Another important mechanism by which RIC can limit free radical oxidation is clearly demonstrated in the model of acetaminophen-induced acute liver injury in mice. In this model, RIC in pre- and post-conditioning modes significantly reduces damage-induced levels of alanine transaminase (ALT), aspartate transaminase (AST), TNFα, IL-6, and MDA in serum, as well as the formation of nitrotyrosine, and stimulates the activity of hepatic SOD, glutathione, and glutathione peroxidase and the expression of HO1 [100]. Anti-oxidant mechanisms of RIC also include activation of G protein-coupled A1 adenosine receptors, which provides neuroprotection by regulating inflammation by lowering serum TNFα and NO levels, enhancing anti-oxidant levels and increasing ATP levels [107].

In experiments with ischemia or the administration of lipopolysaccharides, it has been shown that RIC affects the key steps of systemic inflammation. In particular, it suppresses the activation of nuclear factor NF-κB, significantly reduces the levels of IL-1, IL-6, TNF-α, and IFN-γ both in plasma and in the brain, and suppresses expression of ICAM-1 and VCAM-1 adhesion genes [108]. It also modulates the expression of hypoxia-induced factor HIF-1α and markedly increases levels of HO1, which leads to the strengthening of the blood–brain barrier and a decrease in the infiltration of pro-inflammatory immune cells [108]. Inhibition of NF-κB-dependent pro-inflammatory pathways is considered the key event in preventing neuroinflammatory damage. In particular, blocking the release of cytokines can be carried out by reducing the NF-κB-mediated production of the NLRP3 inflammasome. This can be achieved by a characteristic for RIC decrease in plasma of alarmin HMGB1, which triggers the secretion of pro-inflammatory cytokines through the IκB or ERK activation pathways induced by binding of NF-κB to TLR4 or RAGE (receptors for advanced glycation products) [48]. On the other hand, the association of TNFα or HMGB1 with TLR4 via subsequent activation of NF-κB can affect the mitochondrial pore and synthesis of cytosolic HMGB1, i.e., promotes cell survival. RIC also mediates the decrease in cytokines by suppressing the myeloperoxidase (MPO) pathway, which increases the influx of neutrophils into the area of inflammation and, therefore, promotes the release of pro-inflammatory cytokines from them [16,44,94]. 

Ischemia-induced factor Nrf2 and AMPK kinase may also promote the transition of brain microglial immune cells to the anti-inflammatory M2 phenotype, as opposed to pro-inflammatory M1 phenotype [109]. Damaged neurons release ATP and UTP, which act on purine receptors, causing microglia to differentiate towards the M1 phenotype. M1 microglia promote inflammation and disrupt axonal regrowth by releasing pro-inflammatory cytokines and NO, but they also clear cellular debris. In contrast, M2 microglia produce anti-inflammatory cytokines, VEGF, BDNF, platelet-derived growth factor (PDGF) and progranulin, which, together, suppress inflammation and promote axonal proliferation, angiogenesis, oligodendrogenesis, and remyelination. Preventing this phenotypic shift from M2 to M1 may contribute to recovery from stroke [110].

Suppression of inflammatory reactions by intermittent ischemia is also mediated by chemokines, in particular, by a decrease in the content of monocyte chemoattractant protein-1 (MCP-1) and, subsequently, the intensity of selective recruitment of monocytes [52,67]. In a model of spinal cord injury in rodents, the administration of a RIC-activated humoral factor, stromal cell-derived factor 1α (SDF-1α), reduced the production of inflammasomal IL-1, IL-18, TNFα and NLRP3, confirming that RIC has anti-inflammatory effects [111]. Endocannabinoids also mediate the protective effect of ischemic conditioning through their CB1/CB2 receptors, whose activation causes a decrease in the formation of ROS, chemotaxis, and activation of inflammatory cells, as well as a decrease in internal body temperature, and an increase in coronary and cerebral dilatation, which prevent post-stroke motility disorders [112]. 

For the role of other potential RIC mediators, it is worth considering prostaglandins, which, in acute and chronic neurological conditions, might have protective or toxic properties. It was reported that inhibition of the G-protein-bound prostaglandin F receptor of PGF2α in the CNS via attenuating intracellular calcium levels, improves neurobehavioral function and reduces infarct volume in mice after ischemia [113]. On the other hand, forearm ischemic preconditioning in humans was shown to increase venous plasma prostacyclin levels and arterial plasma levels of BDNF and VEGF improving microvascular endothelial function [114].

In animal models of cerebral ischemia, RIC was reported to cause an increase in the volume of the spleen, the percentage of cytotoxic T cells in it, the number of circulating B lymphocytes and the number of colonies of non-inflammatory monocytes (CD43+/CD172a+). At the same time, a decrease in the content of cytotoxic T cells and natural killer cells was observed in the brain. It is assumed that in the splenic axis of RIC protection, an important role can be played by the markedly increased levels of anti-inflammatory cytokine IL-10, which regulates the amplitude of the cytokine response [52] (Figure 1).

## 7. Role of HIF-1α and Steroid Hormones in RIC

Preclinical studies have proven that activation of the HIF-1α pathway, a transcription factor that plays an important role in response to hypoxia and regulation of inflammation, energy metabolism, neurogenesis, and apoptosis, plays a crucial role for the neuroprotective effects of RIC [115]. The use of a remote ischemic stimulus was shown to significantly increase HIF-1α mRNA and protein levels, leading to reduced cerebral damage, whereas administration of HIF-1α antagonists eliminated the neuroprotective effect of RIC [16]. Inactivation of the HIF-1α subunit expression leads to increased brain damage and decreased survival after ischemia and to a more pronounced learning disorder and decreased neurogenesis in the post-ischemic period [116]. 

In addition to hypoxia, HIF-1α can be induced by NF-κB, growth factors (IGF-1, PDGF), cytokines (TNFα and IL-1), and ROS. HIF-1α controls expression of more than 700 different target genes that mediate both adaptive and pathological processes. In neurons and astrocytes, it controls production of the protective cytokine erythropoietin which regulates apoptosis and autophagy, synaptic processes, and neurogenesis, as well as the inflammatory response, by reducing the expression of cyclo-oxygenase 2 and iNOS and suppressing microglial activation [117,118]. Figure 2 presents some examples of complex interplay between HIF-1α-dependent mechanisms by which RIC promotes neuronal cell survival and reduces damaging effects of ischemia and stroke.

HIF-1α stimulates the production of the vascular growth factor VEGF which reduces the content of active caspase-3 in the hippocampus, modulates the expression of genes that are involved in glucose metabolism, for example, glucose transporter-1 (*GLUT1*) and lactate dehydrogenase A (*LDH-A*) [119]. HIF-1α at the transcriptional level induces the expression of hypoxia-sensitive microRNAs that regulate the stability and translation of mRNA by binding to a 3′ non-coding region, resulting in degradation of the target mRNA and decreased levels of the corresponding proteins [120]. HIF-1α subunits are involved in the regulation of anti-apoptotic factors Bcl-2 and Mcl-1, the induction of Bcl-xL and the suppression of pro-apoptotic factors Bid, Bax, and Bak [121]. HIF-1α is also associated with the regulation of mitochondrial functions, including the control of hexokinase II expression, which catalyzes the first stage of glycolysis and can suppress apoptosis [122]. 

The role of HIF in stroke remains controversial because it is highly associated with the duration and severity of ischemia and as such might activate both protective and pathogenic mechanisms which requires different therapeutic strategies [117,123]. For example, HIF-1 during the early acute phase of the hypoxic response triggers a cascade of cerebral events associated with the suppression of the pentose phosphate pathway [124]. HIF-1α can also trigger p53-induced apoptosis through direct protein interaction of the oxygen-dependent degradation domain of HIF-1α with p53, stabilizing the latter, and through interaction with Mdm2, the modulator of p53 function [125,126]. 

The effect of HIF-1 on the induction of apoptosis also depends on the severity of hypoxia. Under moderate hypoxia or brief ischemia, HIF-1 has a predominantly protective effect by inducing the expression of anti-apoptotic proteins. During the periods of low oxygen pressure, HIF-1α mediates a shift in mitochondrial metabolism towards anaerobic glycolysis, which induces the production of pyruvate dehydrogenase 1 kinase (PDK1) and restricts the entry of acetyl-CoA into the tricarboxylic acid cycle. However, in differentiated macrophages, this metabolic change leads to increased synthesis of cytokines, such as IL-1β and IL-18, via the NF-κB pathway [115,118,126]. HIF-1α also can modulate the activity of NF-κB participating in the regulation of the PI3K/Akt pathway. It is possible that repeated stimulation of HIF-1α causes a separation of cytokine synthesis and immune tolerance, as with other TLR4-dependent pathways [108,117,127].

HIF-1α is an essential component of the pathways controlling cellular metabolism and plays an important role in regulating the effector functions of immune cells. In addition, HIF-1α is crucial for the maturation of dendritic cells and for the activation of T cells. HIF-1α induced in LPS-activated macrophages is crucially involved in glycolysis and induction of pro-inflammatory genes, especially IL-1β [128]. The mechanism of LPS-stimulated induction of HIF-1α involves succinate, which inhibits prolyl hydroxylase and prevents degradation of HIF-1α. Moreover, activated pyruvate kinase M2 interacts with HIF-1α and promotes its function. In another critical type of inflammatory cell, Th17 cells, HIF-1α acts through the retinoic acid-bound orphan receptor-γt to control their differentiation. Thus, HIF-1α acts as a key re-programmer of inflammatory cell metabolism that activates expression of inflammatory genes [129].

In addition to the direct modulation of inflammation, HIF-1α is an important regulator of steroid synthesis. Its expression in adrenal cells dramatically affects the synthesis of hormones with systemic consequences [130]. HIF-1α deficiency causes an increase in the levels of enzymes responsible for steroidogenesis and a corresponding increase in circulating steroids, which leads to changes in cytokine levels and in the profile of circulating mature hematopoietic cells. Overexpression of HIF-1α mediates the insufficiency of steroid production due to impaired transcription of essential enzymes. Such abrupt or sustained changes affect many organ systems, and in particular, sensitive areas of the brain.

The effects of RIC on the regulation of the hypothalamic–pituitary–adrenocortical axis (HPA), which is the main hormonal stress-response system of the organism, are of undoubted theoretical and practical interest, but are currently insufficiently studied. Despite the insufficiency of the experimental data, it is logical to assume the participation of glucocorticoid hormones in the mechanisms of RIC for several reasons. First, there is a significant cross-interaction between the effects of hypoxia/ischemia and glucocorticoids (GCs) on homeostasis and the regulation of cellular responses to oxygen deficiency, stress, and inflammation [131,132]. Glucocorticoid receptors (GRs) and HIFs, whose participation in the effects of RIC has been unequivocally proven, can be colocalized in the same compartments of the nucleus, and probably interact directly. For example, in juvenile zebrafish, HIF-1α suppresses the GR response to exogenous glucocorticoids and reduces cortisol levels by inhibiting proopiomelanocortin (POMC) expression and blocking intracellular GR transcriptional activity. GCs, by contrast, stabilize HIF through pVHL degradation [133]. The functional role of HIF-1α in the regulation of GR mRNA and protein expression, and the associated GC activity, has been demonstrated, and, conversely, HIF-dependent gene expression is enhanced by ligand-dependent activation of GRs (for review, see [134]. It is reasonable to expect that such interplay between HIF-1α and GRs takes plays in the effects of RIC (Figure 2).

HIF and GCs exert a direct and cell-type-specific effect on each other, enhancing or suppressing the transcription of the *N3RC1* and *HIF* genes, respectively, since the promoter of the *N3RC1* gene contains the HRE region and the promoter of the *HIF-1α* GRE element [135]. HIF and GRs can compete for promoters of effector proteins, for example, in the pulmonary epithelium, where the decrease in the anti-inflammatory effect of GCs in hypoxia can be explained by the binding of HIF-1α to HRE present in the promoter of histone deacetylase 2 (HDAC2), which is usually recruited by activated GRs to suppress NF-κB-mediated transcription of pro-inflammatory proteins [136]. These factors can act synergistically. For example, hypoxia increases the expression of the GC-inducible protein GILZ (glucocorticoid-induced leucine zipper) common in the cells of the immune system, which can suppress the activation of macrophages, NF-κB-dependent production of pro-inflammatory cytokines, and inflammatory mediators. In addition, HIF-1α can physically interact with GILZ, which explains the suppression of hypoxia-induced expression of COX-2 by a synthetic GC, dexamethasone [137]. Administration of dexamethasone weakens the activity of HIF-1α in hypoxia by reducing its binding to DNA and HRE activity due to the difficulty of nuclear translocation of HIF-1α. Hypoxia, in turn, can cause a decrease in the levels of GR mRNA and protein and inhibit nuclear translocation of GR, which weakens the anti-inflammatory effect of dexamethasone [131,132,138].

An argument in favor of involving HPA in the central effects of RIC can be the fact that GC has a certain neuroprotective effect, weakening the inflammatory response in ischemic damage due to a significant decrease in the production of TNFα, inhibition of cleaved caspase-3, activation of phosphorylated Akt, and its effect on the VEGF pathway. Dexamethasone and prednisolone are routinely used to treat asthma, ischemic lesions, and consequences of neonatal hypoxia and to prevent the symptoms of altitude sickness. In these cases, they mainly act as anti-inflammatory drugs but also reduce the permeability and vasoconstriction, improving blood oxygenation and redox balance [131]. In our recent experiments on pharmacological preconditioning with dexamethasone, a significant hypoxia-protective effect of GCs was also demonstrated [139]. We have shown a significant effect of hypoxia, especially after conditioning by moderate hypoxia, on the function of HPA and on the number of GR in the brain of experimental animals [32,140].

According to current views [20,23], excess of corticosteroids after focal brain damage via interaction with GRs of the hippocampus, especially in patients with dysregulation of HPA and an abnormal stress response, causes molecular, functional, and structural changes, leading to cognitive and mental disorders. Limb ischemia/reperfusion, being a mild stressor, and having some mechanisms common to conditioning effects, can attenuate the level of steroid hormones and the function of HPA. In our experiments with modeling depression and PTSD in rats, RIC normalized the basal level of GCs in the blood of animals and modulated the HPA, preventing its dysregulation by a feedback mechanism [30,32]. With a high probability, RIC is also able to positively affect the functioning of HPA under conditions of hypoxia, ischemia, and brain damage, preventing excessive secretion of corticosteroid hormones, overactivation of GCs in the hippocampus and its “remote” damage, which undoubtedly deserves further investigation. Particularly interesting results are expected when RIC is applied for the prevention and correction of cognitive and depressive disorders after stroke or ischemic attacks against the background of previous dysregulation of HPA. 

Today, the use of hypoxic conditioning of various modalities that leads to activation of HIF-1α-related cyto-protective mechanisms is also actively discussed in the context of reduction of the severity of the symptoms in COVID-19 patients [123].

## 8. Conclusions

Rapidly accumulating evidence clearly demonstrates that remote ischemic conditioning is a powerful tool for preventing brain damage after stroke or traumas of various nature as well as against progression of chronic pathologies. Despite some limitation in our knowledge of the deep molecular mechanisms of RIC action at the level of different organs, including the brain, there is still scope for further research in this area of neurology. It is certain that RIC has positive effects on neural repair via activation of neurogenesis, regeneration of axons and dendritic networks, synaptogenesis, myelination, angiogenesis, and BBB functions.

Although existing data are sometimes contradictory because of the differences in the animal models and protocols of RIC application, they pave the path towards further development of the techniques suitable for humans. In our opinion, the post-conditioning mode would be the most applicable measure in such acute conditions as heart attack or stroke, while the preconditioning modes can be used not only in a clinical setting but also for increasing organism tolerance to various types of physical loads (e.g., in sport) and for stress management.

Further studies of the mechanisms of implementation of the adaptive effects of remote ischemic conditioning on the brain and other tissues will expand the range of clinical applications of this technique.

## Figures and Tables

**Figure 1 ijms-24-17032-f001:**
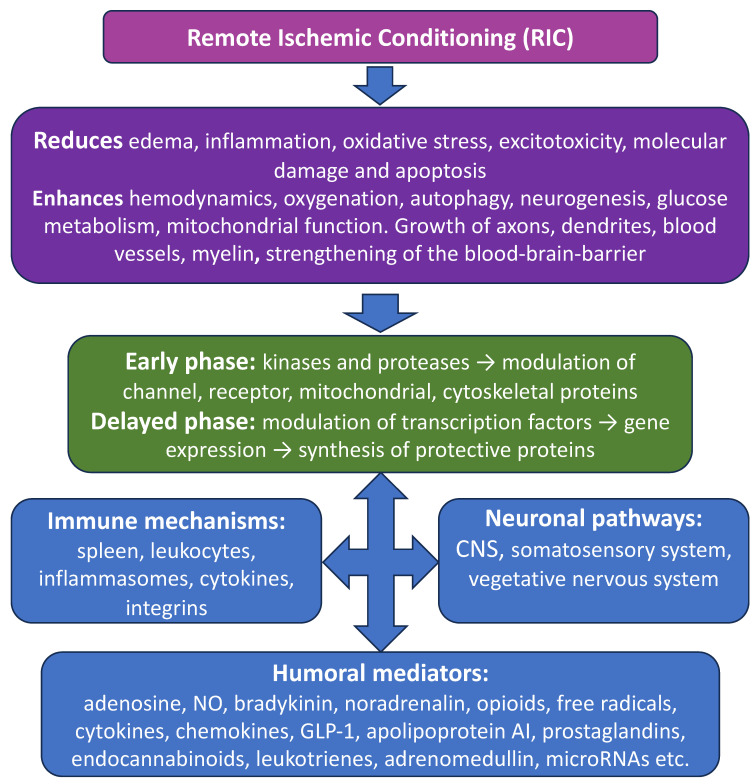
General scheme explaining major events for triggering and implementing the neuroprotective effects of remote ischemic conditioning. At the level of the whole brain (**top panel**), RIC reduces edema, inflammation, oxidative stress, and excitotoxicity, which leads to reduced molecular damage and apoptotic cell death. It also enhances hemodynamics and microcirculation, oxygenation, autophagy, glucose metabolism, and mitochondrial function, which protect neuronal cells and lead to neurogenesis, growth of axons, dendrites, blood vessels, myelin, and strengthening of the blood–brain barrier. These RIC effects have two phases of action at the molecular level. The first early phase involves kinases and proteases, which modulate membrane channels and receptors and affect mitochondrial and cytoskeletal proteins. The second, delayed phase, involves modulation of transcription factors and subsequent increase in gene expression leading to enhanced synthesis of protective proteins. All the events described in the two top panels are interlinked via the immune mechanisms involving spleen, leukocytes, inflammasomes, cytokines, integrins, etc., and neuronal pathways (CNS, somatosensory system, vegetative nervous system). These interactions are facilitated by a range of humoral mediators (**bottom panel**).

**Figure 2 ijms-24-17032-f002:**
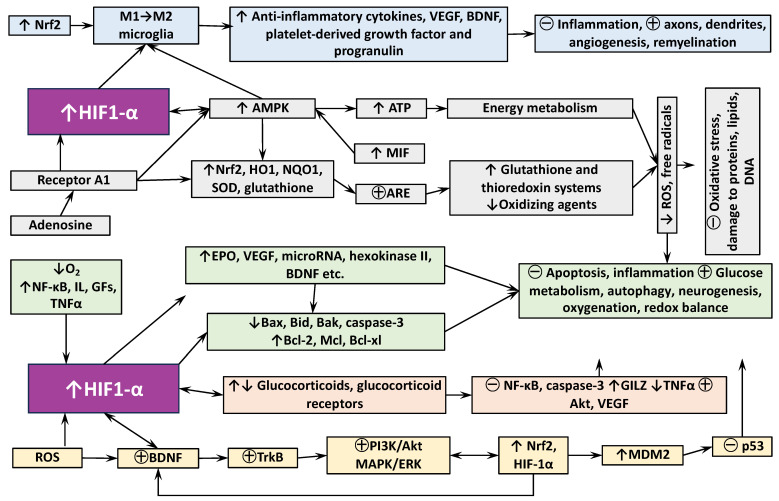
Some examples of HIF-dependent neuroadaptive mechanisms of remote ischemic conditioning demonstrating complex interplay of various factors involved in neuronal cell survival and death. Abbreviations: HIF1-α—hypoxia-inducible factor-α; Nrf2—the nuclear factor erythroid; M1→M2—shift from proinflammatory microglia M1 type to tissue repairing M2 microglia; VEGF—vascular endothelial growth factor; BDNF—brain-derived neurotrophic factor; AMPK—adenosine monophosphate-activated protein kinase; ATP—adenosine triphosphate; ROS—reactive oxygen species; Receptor A1—G protein-coupled A1 adenosine receptors; HO1—hemoxygenase-1; NQO1—quinone oxidoreductase 1; SOD—superoxide dismutase; MIF—migration inhibiting factor; ARE—anti-oxidant response element; NF-κB—nuclear factor κ-light-chain-enhancer of activated B cells; IL—interleukins, GFs—growth factors; TNFα—tumor necrosis factor-α; EPO—erythropoietin; Bax, Bid, Bak, Bcl-xl—members of Bcl-2 gene family regulating apoptosis; GILZ—glucocorticoid-induced leucine zipper (GC-inducible protein); Akt—protein-kinase type B; TrkB—tropomyosin receptor kinase B; PI3K—phosphoinositide 3-kinase; MAPK—mitogen-activated protein-kinase; ERK—extracellular signal-regulated kinases; MDM2—mouse double minute 2 homolog (also known as E3 ubiquitin-protein ligases Mdm2—negative regulator of tumor suppressor protein p53. ↑—increase or ↓—decrease in expression, synthesis, or activity; ㊉—activation, strengthening or ㊀—inhibition, weakening of the processes.

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
