# Peer review of "Neuroadaptive Biochemical Mechanisms of Remote Ischemic Conditioning"

_ijms, 2023, doi:10.3390/ijms242317032_

Round 1
Reviewer 1 Report
Comments and Suggestions for Authors
The manuscript, entitled Neuroadaptive Biochemical Mechanisms of Remote Ischemic Conditioning is a high-quality summary publication work, based on a total of 127 up-to-date references.
I have the following comments and suggestions regarding the manuscript:
- although the manuscript is very well edited and discusses the remote ischemic conditioning neuroadaptive biochemical mechanism in detail, however, I miss the discussion, even in a short form, of the effects of microcirculation on the referred topic
- in line 17., the abbreviation HIF must be explained
- in line 408., the subsection Role of HIF-1α and steroid hormones is correctly numbered 7 and not 6.
- in line 497., in the words N3RC1 and HIF the characters are unnecessarily displayed in italic type
Author Response
Reviewer 1
Comments and Suggestions for Authors
The manuscript, entitled Neuroadaptive Biochemical Mechanisms of Remote Ischemic Conditioning is a high-quality summary publication work, based on a total of 127 up-to-date references.
Many thanks for your encouraging comments about our manuscript.
I have the following comments and suggestions regarding the manuscript:
They are answered below:
- although the manuscript is very well edited and discusses the remote ischemic conditioning neuroadaptive biochemical mechanism in detail, however, I miss the discussion, even in a short form, of the effects of microcirculation on the referred topic
We agree that microcirculation plays an important role in the response of the brain and other organs to distant hypoxic/ischemic challenge. We have added some text discussing it and several references at the end of Chapter 3.
- in line 17., the abbreviation HIF must be explained
Abbreviation for HIF has been added.
- in line 408., the subsection Role of HIF-1α and steroid hormones is correctly numbered 7 and not 6.
This has been corrected.
- in line 497., in the words N3RC1 and HIF the characters are unnecessarily displayed in italic type
This has been corrected.
Reviewer 2 Report
Comments and Suggestions for Authors
Authors present an interesting review of mechanisms of remote preconditioning. Overall, article is quite well written. However, there are some major and minor issues.
Major.
1. Sections 5, 6, and 7 can be better organized and feel meandering. I felt quite lost in the middle of the section.
2. Authors need to reference the existing figures within the text and properly describe them.
3. Current Figure 2 covers section 7 (HIF1). Consider adding more figures to illustrate other sections as well.
Minor
1. I believe authors use word distancing in place of distant or remote in some instances.
2. There are two sections #6 and all subsequent sections are numbered incorrectly as a result.
3. Figure 2 has "ATФ". It should be "ATP"
Comments on the Quality of English LanguageLanguage is quite good. I noticed only couple incorrect usage of a single word and one typo.
Author Response
Reviewer 2
Comments and Suggestions for Authors
Authors present an interesting review of mechanisms of remote preconditioning. Overall, article is quite well written. However, there are some major and minor issues.
Thank you for your positive evaluation of the manuscript. We have answered your criticisms as explained below.
Major.
- Sections 5, 6, and 7 can be better organized and feel meandering. I felt quite lost in the middle of the section.
We have edited the text of these sections for clarity and hope it is now more logically presented.
- Authors need to reference the existing figures within the text and properly describe them.
The references for Figures have been added to the text where appropriate and the Legends extended.
- Current Figure 2 covers section 7 (HIF1). Consider adding more figures to illustrate other sections as well.
Many thanks for this suggestion. We think that Figure 1 covers in general all sections of the MS and as such feel that more extensive figures for each section might be redundant.
Minor
- I believe authors use word distancing in place of distant or remote in some instances.
The word “distancing” in two places has been replaced by “remote”.
- There are two sections #6 and all subsequent sections are numbered incorrectly as a result.
This has been corrected.
- Figure 2 has "ATФ". It should be "ATP"
This has been corrected.
Comments on the Quality of English Language
Language is quite good. I noticed only couple incorrect usage of a single word and one typo.
We have done extensive proofreading of the manuscript and corrected the typos.
Reviewer 3 Report
Comments and Suggestions for Authors
In this review authors summarized the main findings regarding the biochemical mechanisms that mediate the neuroadaptive effects of the remote Ischemic conditioning (RIC). Although it is a well-written review, there are some weaknesses, presented below.
-The aim of the paper is not clearly stated in the introduction. In addition, a brief description of the main topics to be presented should have been included.
- I assume that this is a narrative review. In this case, authors should have presented the methodology applied. For example, they could have reported if their review conforms to the SANRA (Scale for the Assessment of Narrative Review) guidelines by reporting also the databases that were searched, the keywords, the total number of articles that were retrieved and the method of screening.
-l. 17: add "those" ("those mediated")
-l. 17: I assume that HIF refers to "Hypoxia-inducible factor". Please use the full words.
-l. 20-21: The full words should be reported instead of HIF-1 and HPA.
-l. 42: please add "also" so that it reads "but also causes".
-l. 60" Do you mean "on other brain regions"?
-l. 73: "in these areas..." since they refer to both CA1 and CA2 hippocampal areas.
-L. 57-76: Please rephrase.
-l. 100: "hippocampal" behavioral outcome: please elaborate on this. What do you mean?
-l. 300: "The prevention....microautophagy": please rephrase.
-The tiles of the different sections should be more specific. In other words, they should reflect/summarize the content of each section. For example, titles of sections 3-6 are too general, and not linked to the RIC.
-l. 408: "6" should change to "7" and in l. 561 "7" to "8".
-l. 562-569: Prior to the conclusions, a short discussion of the limitations of the existing evidence should have been included. In such a section, authors could also refer to future directions in this research field.
Author Response
Comments and Suggestions for Authors
In this review authors summarized the main findings regarding the biochemical mechanisms that mediate the neuroadaptive effects of the remote Ischemic conditioning (RIC). Although it is a well-written review, there are some weaknesses, presented below.
-The aim of the paper is not clearly stated in the introduction. In addition, a brief description of the main topics to be presented should have been included.
This has been addressed in the Introduction section.
- I assume that this is a narrative review. In this case, authors should have presented the methodology applied. For example, they could have reported if their review conforms to the SANRA (Scale for the Assessment of Narrative Review) guidelines by reporting also the databases that were searched, the keywords, the total number of articles that were retrieved and the method of screening.
This review paper was written based on the authors’ expertise and knowledge in the field. All cited papers have been read and analyzed by the authors in the process of creating a model of remote ischemic conditioning in the laboratory. The papers have been retrieved via PubMed or other internet-based search engines and no software was used for narrating the review. We do not consider it as a “narrative” medical review and as such SANRA has not been used. We have now emphasized in the Introduction the purpose of this review and implementation of our intellectual know-how in the process of writing it.
-l. 17: add "those" ("those mediated")
Added.
-l. 17: I assume that HIF refers to "Hypoxia-inducible factor". Please use the full words.
Done.
-l. 20-21: The full words should be reported instead of HIF-1 and HPA.
Done.
-l. 42: please add "also" so that it reads "but also causes".
Done.
-l. 60" Do you mean "on other brain regions"?
Correct. The word “brain” is added for clarity.
-l. 73: "in these areas..." since they refer to both CA1 and CA2 hippocampal areas.
Corrected.
-L. 57-76: Please rephrase.
We have slightly edited this section which we believe has improved the clarity. See lines 57-72.
-l. 100: "hippocampal" behavioral outcome: please elaborate on this. What do you mean?
This has been edited for clarity.
-l. 300: "The prevention....microautophagy": please rephrase.
Done.
-The tiles of the different sections should be more specific. In other words, they should reflect/summarize the content of each section. For example, titles of sections 3-6 are too general, and not linked to the RIC.
The titles of the sections have been extended for clarity.
-l. 408: "6" should change to "7" and in l. 561 "7" to "8".
Corrected.
-l. 562-569: Prior to the conclusions, a short discussion of the limitations of the existing evidence should have been included. In such a section, authors could also refer to future directions in this research field.
This has been added.
Round 2
Reviewer 3 Report
Comments and Suggestions for Authors
Authors have addressed my comments. However, the way they responded is not appropriate (in most cases their reply included just a word like "Added", "Corrected" "Done"). They had to explain how they addressed each comment. In addition, the authors had to include the exact lines in which the changes appear in the revised document.